# Identification of Pri-miRNA Stem-Loop Interacting Proteins in Plants Using a Modified Version of the Csy4 CRISPR Endonuclease

**DOI:** 10.3390/ijms23168961

**Published:** 2022-08-11

**Authors:** Janina Lüders, Andreas R. Winkel, Marlene Reichel, Valentin W. Bitterer, Marion Scheibe, Christiane Widmann, Falk Butter, Tino Köster

**Affiliations:** 1Department of RNA Biology and Molecular Physiology, Faculty of Biology, Bielefeld University, D-33615 Bielefeld, Germany; 2Quantitative Proteomics, Institute of Molecular Biology (IMB), D-55128 Mainz, Germany; 3Structural Biochemistry, Department of Chemistry, Bielefeld University, D-33615 Bielefeld, Germany

**Keywords:** RNA-binding protein, pri-miRNA stem-loop, RNA-protein interaction, RNA pulldown, Csy4* CRISPR endonuclease, Arabidopsis, plant

## Abstract

Regulation at the RNA level by RNA-binding proteins (RBPs) and microRNAs (miRNAs) is key to coordinating eukaryotic gene expression. In plants, the importance of miRNAs is highlighted by severe developmental defects in mutants impaired in miRNA biogenesis. MiRNAs are processed from long primary-microRNAs (pri-miRNAs) with internal stem-loop structures by endonucleolytic cleavage. The highly structured stem-loops constitute the basis for the extensive regulation of miRNA biogenesis through interaction with RBPs. However, *trans*-acting regulators of the biogenesis of specific miRNAs are largely unknown in plants. Therefore, we exploit an RNA-centric approach based on modified versions of the conditional CRISPR nuclease Csy4* to pull down interactors of the Arabidopsis pri-miR398b stem-loop (pri-miR398b-SL) in vitro. We designed three epitope-tagged versions of the inactive Csy4* for the immobilization of the protein together with the pri-miR398b-SL bait on high affinity matrices. After incubation with nucleoplasmic extracts from Arabidopsis and extensive washing, pri-miR398b-SL, along with its specifically bound proteins, were released by re-activating the cleavage activity of the Csy4* upon the addition of imidazole. Co-purified proteins were identified via quantitative mass spectrometry and data sets were compared. In total, we identified more than 400 different proteins, of which 180 are co-purified in at least two out of three independent Csy4*-based RNA pulldowns. Among those, the glycine-rich RNA-binding protein *At*RZ-1a was identified in all pulldowns. To analyze the role of *At*RZ-1a in miRNA biogenesis, we determined the miR398 expression level in the *atrz-1a* mutant. Indeed, the absence of *At*RZ-1a caused a decrease in the steady-state level of mature miR398 with a concomitant reduction in pri-miR398b levels. Overall, we show that our modified Csy4*-based RNA pulldown strategy is suitable to identify new *trans*-acting regulators of miRNA biogenesis and provides new insights into the post-transcriptional regulation of miRNA processing by plant RBPs.

## 1. Introduction

MicroRNAs (miRNAs) are small 21–24 nucleotide (nt) long noncoding RNAs that are considered as a fundamental hub within the regulation of eukaryotic gene expression [1,2,3]. The importance of miRNAs in plants is underscored by the association of miRNA dysregulation with disturbances in fundamental biological processes, such as development, hormone regulation, and responses to various abiotic and biotic stresses [4,5]. The biogenesis of miRNAs is a multistep process. After transcription by RNA polymerase II, the conversion of long stem-loop-containing primary transcripts (pri-miRNAs) to shorter stem-loop precursor-miRNAs (pre-miRNA), and further processing to the mature miRNAs takes place in the nucleus. Both processing steps are executed by the same enzyme. Of the four RNase III-type DICER-LIKE (DCL) proteins, DCL1 is the main enzyme involved in miRNA biogenesis, generating predominantly miRNAs of 21 nt. The mature miRNAs are loaded into ARGONAUTE (AGO) effector proteins. The resulting RNA-Induced silencing complex (RISC) is recruited to miRNA target sites and downregulates the gene expression by the cleavage of the target mRNA or by translational repression [6,7].

MiRNA biogenesis in plants involves the dynamic interaction of several RNA-binding proteins (RBPs) with pri-miRNAs. DCL1 interacts with the double-stranded RBP HYPONASTIC LEAVES 1 (HYL1), the zinc finger protein SERRATE (SE), and the G-patch domain protein TOUGH (TGH) that contribute to the efficiency and/or accuracy of pri-miRNA processing [8,9,10]. DAWDLE (DDL), a forkhead-associated (FHA) domain protein, is assumed to recruit DCL1 to pri-miRNAs and stabilize pri-miRNAs [11,12]. Furthermore, the nuclear cap binding proteins CBP20 and CBP80 that interact with the cap structure at the 5′end of the pri-miRNA affect the biogenesis of a set of miRNAs [13,14,15].

Importantly, the stem-loops of plant pri-miRNAs are highly variable in length and structure, even within the same miRNA families. These structural properties likely constitute a basis for the extensive post-transcriptional regulation of pri-miRNA processing through the direct binding of RBPs. Moreover, discrepancies between pri-miRNA and mature miRNA levels underscore the importance of post-transcriptional control in miRNA biogenesis [16].

Although several RBPs have been implicated in the regulation of miRNA biogenesis, the knowledge of the *trans*-acting regulators of the biogenesis of specific miRNAs is limited, especially in plants [1,17]. A number of different strategies have been developed to isolate RNA-associated factors in specific ribonucleoprotein complexes [18]. In RNA-centric approaches, the RNA is biotinylated or tagged with a specific recognition sequence. RNA–protein complexes are assembled by incubating the in vitro-transcribed RNA with cell lysates and are affinity-enriched under native conditions. Bound proteins are eluted from the RNA and subjected to mass spectrometry (MS). Some strategies rely on aptamers, e.g., short sequences derived from viral heterogenous RNA stem-loops selected for high affinity binding to other molecules, such as MS2 or PP7 [19,20].

An alternative method for the identification of RBPs that specifically associate with RNAs is based on an engineered version of the clustered regularly interspaced short palindromic repeats (CRISPR) Csy4 endoribonuclease from *Pseudomonas aeruginosa* [21]. Csy4 selectively recognizes a 16-nt hairpin sequence encoded by the CRISPR repeat with a K_d_ of 50 pM and cleaves at the bottom of this stable hairpin [22]. The mutation of histidine 29 to alanine in the active site leads to an enzymatically inactive Csy4^H29A^ that still recognizes the hairpin [21,23]. To capture the proteins binding to stem-loops of mammalian pre-miRNAs, corresponding in vitro transcripts were tagged with the Csy4 hairpin at their 5′end. An additional mutation of serine 50 on the protein surface to cysteine (Csy4^H29AS50C^, Csy4*) allows for biotin labeling and immobilizing on an avidin resin. The inactive and biotinylated Csy4* binds to the tagged pre-miRNA of interest and interacting proteins were precipitated from cell lysates by immobilized Csy4*–pre-miRNA complexes. An addition of imidazole recovered the Csy4* cleavage activity, removing the tag and leading to the specific elution of the pre-miRNA–protein complexes. The co-precipitating proteins were then identified via MS. Using this system, Lee et al., have identified the proteins interacting with different pre-miRNAs in HeLa and NTERA2 cell extracts [21].

The key to a successful Csy4*-based RNA pulldown is the recombinant expression and purification of the tagged and functional His_6_-MBP-Csy4* fusion protein [21,23]. In the original protocol, recombinant His_6_-MBP-Csy4* is expressed in *Escherichia coli* and is initially purified via Ni-NTA resin in batch. Protein eluted with imidazole is then treated with tobacco etch virus (TEV) protease to release the His_6_-MBP tag that is then removed by a second nickel-affinity step. The protein is further purified on a preparative scale by a size exclusion chromatography system. For immobilization, the purified Csy4* protein is biotinylated. This multistep strategy is technically demanding and requires extensive dedicated instrumentation.

Therefore, our aim was to develop a simplified version of the original Csy4* protocol, which is easily applicable for routine use in molecular biology laboratories. For this reason, we have selected and engineered three different epitope-tagged versions of Csy4* and compared their performance in RNA pulldown experiments. As an example, we selected the stem-loop structure of pri-miR398b, a member of the evolutionary conserved and stress-responsive miR398 family. Specific protein interactors were identified by MS in three independent Csy4*-based pulldowns. Among those, we found the glycine-rich protein *At*RZ-1a. For functional validation, we investigated the influence of *At*RZ-1a on the processing of pri-miR398b into mature miR398 in the loss-of-function mutant *atrz-1a.* Our results show decreased steady-state levels of both mature miR398 and its precursor pri-miR398b in the *atrz-1a* mutant. This suggests that our modified Csy4*-based strategy provides a valuable method for the identification of specific regulatory proteins in miRNA biogenesis.

## 2. Results and Discussion

An increasing number of RBPs has been recently described to be involved in plant miRNA biogenesis [17]. The various functions of newly identified RBPs and their regulatory involvement in the miRNA biogenesis pathway suggest that the regulation of pri-miRNA processing is highly complex and more dynamic than previously thought. Although several RBPs have been shown to be involved in this process, the knowledge about specific regulatory proteins recognizing particular miRNA precursors is still limited. In vitro RNA-affinity purification methods allow for the simple and rapid identification of proteins that interact with particular pri-miRNA stem-loops. A key challenge in RNA affinity purifications is to distinguish between specifically bound proteins and unspecific background noise. Recently, Lee at al. developed a CRISPR-based strategy for the identification of RNA–protein complexes to overcome this problem [21]. Here, we applied this RNA-centric approach based on the engineered CRISPR Csy4* endoribonuclease to plants, where we used nucleoplasmic extracts to identify novel interactors of the pri-miR398b stem-loop (-SL) structure (Figure 1A).

### 2.1. Experimental Strategy and Recombinant Expression of Tagged Csy4* Protein Variants

As the recombinant expression and purification of the functional Csy4* is technically demanding and requires extensive dedicated instrumentation, we developed modified versions of the original Csy4*-based strategy to identify the interacting proteins of the pri-miRNA stem-loops in nucleoplasmic extracts from *Arabidopsis thaliana*. To do this, we applied three different tagged versions of the Csy4* protein. All epitopes were selected based on the criteria of easy and inexpensive protein preparation, high selectivity and the availability of high-affinity matrices for immobilization. Large-size tags such as MBP can have a positive effect on protein solubility and expression efficiency [24,25]. For this reason, and due to the availability of specific camelid heavy chain anti-MBP-tag nanobodies which show an exceptionally high affinity towards the MBP-tag in the nM range, we initially decided to purify the original His_6_-MBP-Csy4* protein [21]. In contrast to conventional antibodies, nanobodies are devoid of light chains and bind their antigen via a single variable domain (V_H_V) [26]. Previously, nanobodies have been successfully used for protein-centric methods of identifying targets of RNA-binding proteins with high reproducibility [27,28,29]. In contrast to large tags such as MBP, small-size tags generally exert minimal effects on the structure and biological activity of their fused proteins. Therefore, we replaced the large-size MBP-tag with two smaller tags, i.e., the Myc-tag and the TwinStrep-tag, respectively. The Myc-tag is a 1.2 kDa peptide tag containing 10 amino acids (EQKLISEEDL) derived from the human c-Myc oncogene [30]. As for the MBP-tag, anti-Myc nanobodies (Myc-Trap^®^) can be used for the immobilization of the His_6_-Myc-Csy4* fusion protein. The TwinStrep-tag exploits one of the strongest non-covalent interactions in nature and is originally based on the interaction between biotin and avidin [31,32]. It consists of two eight amino acid repeats (WSHPQFEK) connected by a twelve amino acid linker (GGGS_2_-GGSA) and exhibits very high affinity in the pM range to the streptavidin derivative Strep-Tactin^®^ XT.

For the recombinant expression of His_6_-MBP-Csy4* in *E. coli*, we used the original Gateway-based pHM-GWA_Csy4H29AS50C expression vector (Appendix A) [21,33]. The expression of the 64.9 kDa His_6_-MBP-Csy4* protein is efficiently induced by the addition of IPTG and the fusion protein is purified via Ni-NTA affinity resin with a high yield (Appendix A). To produce the 25.4 kDa His_6_-TwinStrep- and 23.6 kDa His_6_-Myc-tagged Csy4*, respectively, we replaced the His_6_-MBP coding sequence in pHM-GWA_Csy4H29AS50C (Appendix A) with the sequences for His_6_-TwinStrep (Appendix A) and His_6_-Myc (Appendix A). We again tested the expression and purification efficiency of both Csy4* protein variants. Additionally, we added further washing steps with increasing imidazole concentrations to remove unspecific proteins during purification (Appendix A). Furthermore, we determined the optimal buffer conditions for the less soluble His_6_-Myc-tagged Csy4* protein by differential scanning fluorimetry (DSF) in order to avoid the aggregation and precipitation of His_6_-Myc-Csy4* protein during purification [34]. We tested the stability of His_6_-Myc-Csy4* in different buffer systems using variable concentrations of sodium chloride (Appendix A). Phosphate buffer with a pH of 7.5 containing 500 mM sodium chloride turned out to be optimal for the stability of His_6_-Myc-Csy4* protein (Appendix A). His_6_-TwinStrep-Csy4* and His_6_-Myc-Csy4* could be expressed successfully, but not with comparable efficiency to His_6_-MBP-Csy4* (Appendix A). Consequently, the yield of both His_6_-TwinStrep- and His_6_-Myc-tagged Csy4* protein variants after nickel-affinity purification is lower compared to His_6_-MBP-Csy4*. Overall, sufficient protein of all three tagged Csy4* protein variants could be produced for several RNA pulldowns.

### 2.2. Functional Characterization of Different Tagged Csy4* Proteins

As an example for the identification of pri-miRNA stem-loop interacting proteins, we have chosen the Arabidopsis pri-miR398b-SL structure. Pri-miR398b is a member of the stress-responsive miR398 family which targets two Cu/Zn-dismutases and a copper chaperone [35]. Initially, we designed a 5′ Csy4-hairpin-tagged stem-loop of pri-miR398b for in vitro transcription using T7 RNA polymerase (Appendix A). As the endogenous pri-miR398b is processed in a base-to-loop direction, the stem-loop construct contains the ~15-nt lower stem region which specifies the position of the first DCL1 cleavage site (Appendix A) [36]. To ensure that the 20-nt Csy4 stem-loop does not interfere with the pri-miR398b stem-loop structure, we tested the RNA secondary structure by in silico prediction using the RNAfold web server tool [37]. To obtain correct folding, we introduced a 5-nt and a 6-nt spacer sequence at the 5′ and 3′ end of the pri-miR398b-SL (Appendix A).

After in vitro transcription, we checked the efficiency of RNA coupling to the respective Csy4* proteins (Figure 1B–D). For this, we immobilized the enzymatically inactive His_6_-MBP-Csy4*, His_6_-Myc-Csy4* and His_6_-TwinStrep-Csy4* protein variants on either the MBP-Trap^®^ agarose beads, Myc-Trap^®^ agarose beads or StrepTactin^®^ XT agarose beads, respectively. We then incubated the in vitro transcribed Csy4-hairpin-tagged pri-miR398b-SL with the immobilized Csy4* protein variants to form resin-bound Csy4*–RNA complexes and tested the depletion of the transcript in the supernatant by comparing the input fraction (IN) with the supernatant (SN) on urea-PAGE gels. For the original His_6_-MBP-Csy4* we reached a depletion of ~80% in the supernatant suggesting that the protein efficiently binds the Csy4-hairpin-tagged pri-miR398b-SL (Figure 1B). Compared to His_6_-MBP-Csy4*, we achieved lower coupling efficiencies for both Csy4* variants fused to either His_6_-TwinStrep or His_6_-Myc. In contrast to the His_6_-Myc-Csy4*, for which we achieved a transcript depletion of almost 50% in the supernatant (Figure 1D), only less than 10% of the transcripts were depleted by His_6_-TwinStrep-Csy4* from the supernatant (Figure 1C). This suggests a lower binding affinity of both protein variants, especially that of the His_6_-TwinStrep-Csy4*.

Next, we tested the cleavage and elution efficiencies of the Csy4* variants (Figure 1B–D). These properties significantly influence the enrichment, specificity and thus the identification of the co-purified proteins in the subsequent RNA pulldowns. For this purpose, the three immobilized Csy4* fusion proteins with their loaded RNA baits were incubated with elution buffer containing 500 mM imidazole to reactivate the cleavage activity (E+). Incubation without imidazole served as a negative control (E−). To precisely determine the cleavage efficiency and to check whether RNA remained on the coupled beads, we also eluted the remaining RNA from the beads after removing the supernatant and again loaded it on the urea-PAGE gels (B+ and B−). When both E+ and B+ fractions are considered for His_6_-MBP-Csy4*, the protein exhibits high overall cleavage activity after activation, as the transcript in both lanes is 20 nt shorter after endonucleolytic cleavage (Figure 1B). However, it was not possible to elute the cleaved transcript efficiently from the immobilized His_6_-MBP-Csy4*, as a large part remained on the beads after elution (B+). Surprisingly, since parts of the transcript are already cleaved on the beads without the addition of imidazole in the B− fraction, a non-specific cleavage activity of the original His_6_-MBP-Csy4* protein cannot be excluded. Although His_6_-Myc-Csy4* exhibits a lower binding efficiency towards the transcript, we observed a reasonable amount of eluted pri-miR398b-SL in the E+ fraction with no unspecific cleavage in the E- fraction (Figure 1D). As expected for His_6_-TwinStrep-Csy4*, we obtained only a small amount of the transcript in the eluate, due to the low binding efficiency (Figure 1C). Considering the observed differences in the coupling and elution properties of the three Csy4* variants, we decided to compare their performance in RNA pulldown experiments.

### 2.3. Preparation of Nucleoplasmic-Enriched Extracts

Opposite to metazoans, all steps of the pri-miRNA processing in plants occur in the nucleus [38]. Recent results suggest a scenario where the processing of base-to-loop pri-miRNAs initiates co-transcriptionally on the gene locus and continues post-transcriptionally with the processing of the pre-miRNA in the nucleoplasm [39]. For this reason, and to avoid contamination with the highly abundant DNA-binding proteins coming from the chromatin, we used nucleoplasmic extracts for our RNA pulldown experiments. Five grams of ground 14-day old Columbia-0 wild-type plants were resuspended in hypotonic buffer, swollen cells were homogenized and nuclei were pelleted. After the removal of the cytoplasmic fraction, three volumes of high-salt lysis buffer containing 420 mM sodium chloride were added to the nuclei pellet to release soluble protein from the nucleoplasm without lysing the nuclei. We evaluated the purity of the nucleoplasmic extracts by immunoblot analysis with antibodies against specific markers for the nucleoplasm (SERRATE, SE), the chromatin (HISTONE 3, H3), and the cytoplasm (PHOSPHOENOLPYRUVATE CARBOXYLASE, PEPC) (Appendix A). The presence of SE and the simultaneous absence of H3 and PEPC in our native extracts confirmed that nucleoplasmic proteins were highly enriched by our isolation method.

### 2.4. Proof-of-Principle: Comparative Analysis of Csy4*-Based RNA Pulldown

To allow for the assembly of the RNA–protein complexes, immobilized Csy4*-hairpin-tagged pri-miR398b-SL were incubated with 1 mg nucleoplasmic protein in 2 mL extract under native conditions. The unspecific background was minimized by washing the immobilized RNA–protein complexes three times with buffer containing 100 mM sodium chloride and 0.1% Triton X-100 as a non-ionic detergent. The RNA–protein complexes were specifically recovered from the beads by the addition of 500 mM imidazole, which led to the reactivation of the site-specific cleavage activity of Csy4*, removing the tag and releasing the pri-miR398b-SL and associated proteins with minimal contamination. After the washing and elution of the RNA, the associated proteins were separated by SDS-PAGE and analyzed by mass spectrometry. As a negative control, we used the same resin-bound Csy4*–RNA complexes and experimental setup, but without the addition of imidazole and elution of bound proteins. For all three Csy4*-based pulldowns, four replicates for both eluates and negative controls were analyzed by quantitative MS. Volcano plots in Figure 1E–G show the proteins co-purified with the pri-miR398b stem-loop for His_6_-MBP-Csy4*, His_6_-TwinStrep-Csy4* and His_6_-Myc-Csy4*, respectively.

In total, the three different approaches identified 173, 250 and 281 proteins, respectively, which were classified as either interactors or candidates (Figure 2A, Appendix A). As interactors, we defined proteins, which are positively enriched and pass the stringent statistical criteria (log_2_ fold change ≥ 1.0; *p*-value ≤ 0.05). All other co-purified proteins, which show a positive log_2_ fold change, are defined as candidates. Although candidates do not pass the stringent statistical criteria and cannot clearly be distinguished from the unspecific background, their features are nevertheless examined below, as they likely contain additional interactors that warrant further investigation. Together, this identified 483 different proteins showing a positive log_2_ fold change in the eluates of the pri-miR398b-SL pulldowns. The comparison of the combined interactors and candidates of the three approaches shows substantial overlap with 182 proteins being identified as an interactor and/or candidate in at least two of the three pulldowns (Figure 2B). Moreover, 151 interactors were significantly enriched in at least one Csy4*-based pulldown.

To validate our purification strategy, we first tested whether we could purify proteins known to be involved in miRNA biogenesis. Previously, it was shown that the levels of pri-miR398a, b and c were accumulated at the expense of the mature miRNAs in plants overexpressing the hnRNP-like protein *At*GRP7 [40]. RNA immunoprecipitation revealed that *At*GRP7 directly interacts with these pri-miRNAs in vivo and this interaction is lost upon the mutation of a single conserved arginine residue in the RNA recognition motif (RRM) [41]. For this reason, we first checked if we found *At*GRP7 among the co-purified proteins. Indeed, *At*GRP7 was found among the candidates in both His_6_-TwinStrep-Csy4*- and His_6_-Myc-Csy4*-based RNA pulldowns (Figure 1F,G and Figure 2C). This strengthens our hypothesis that proteins that play a regulatory role in miRNA biogenesis can also be found among the candidates.

Next, we checked for the co-purification of proteins from the processing core complex and found the C2H2-type zinc-finger protein SE among the interactors in the His_6_-TwinStrep-Csy4*-based pulldown and among the candidates in the His_6_-Myc-Csy4*-based pulldown (Figure 1F,G and Figure 2C). SE binds to pri-miRNA stem-loops at single-stranded to double-stranded RNA junctions [42]. It interacts with the processing components DCL1 and HYL1 to ensure the precise and efficient processing of pri-miRNAs and the *se-1* mutants show reduced miRNA levels and an accumulation of pri-miRNAs [43,44]. Furthermore, we found several components of the MOS4-ASSOCIATED COMPLEX (MAC) that are involved in the miRNA biogenesis pathway (Figure 2C) [45]. MAC3A and MAC3B are core subunits of the MAC and both proteins interact with and promote the activity of DCL1. *Mac3a* and *mac3b* double mutants show reduced levels of miRNAs with a simultaneous reduction in pri-miRNA levels. Importantly, MAC3A associates with some pri-miRNAs in vivo. MAC7, a putative RNA helicase, interacts with HYL1 and other members of the MAC, including MAC3B, and the partial loss-of-function mutant *mac7-1* exhibits reduced miRNA and pri-miRNA levels [46]. Besides MAC7, two other MAC subunits, PRL1 and CDC5, were among the candidates in at least one Csy4*-based pulldown and play important roles in pri-miRNA processing. PRL1 positively regulates the biogenesis and accumulation of mature miRNAs by directly associating with pri-miRNAs in vivo [47]. CDC5 interacts with DCL1 and SE and acts as a component of the processing complex [48]. Furthermore, CDC5 helps to recruit RNA polymerase II to the promoter of *MIR* genes. In the His_6_-TwinStrep-Csy4*-based pulldown, we have also identified SMALL1 (SMA1), a homolog of the mammalian DEAD-box pre-mRNA splicing factor Prp28, among the candidates (Figure 2C). SMA1 interacts with the DCL1 complex and enhances the efficiency of miRNA processing. The absence of SMA1 in Arabidopsis causes a reduction of several mature miRNAs and both miR398b and c are slightly reduced in *sma1-1* mutants [49]. With the WD40 repeat-like superfamily protein RECEPTOR FOR ACTIVATED C KINASE 1 A (RACK1A), the major one of three RACK1 genes in Arabidopsis, we found another regulatory factor among the candidates in the His_6_-MBP-Csy4*-based pulldown (Figure 2C). RACK1A was shown to directly interact with SE and mature miRNA levels are significantly reduced in *rack1a,b,c* triple mutants [50]. Additionally, animal RACK1 has also been shown to act in the miRNA pathway [51,52].

Since the loading of mature miRNAs into the RISC is a dynamic process and a fraction of AGO1 congregates with the processing components DCL1 and HYL1, it is not surprising that AGO1 is found among interactors and candidates in all three Csy4*-based pulldown experiments (Figure 2C) [53]. Furthermore, nuclear HYL1 is required for miRNA strand selection and Bologna et al., recently suggested a model in which mature miRNAs are loaded in the nucleus and require AGO1 nucleo-cytosolic shuttling for their translocation [54,55].

For further validation of our mass spectrometry data, we systematically analyzed the molecular identity of the co-purified proteins by gene ontology (GO) term analysis. Consistent with a putative regulatory function in miRNA biogenesis, co-purified proteins of all three Csy4*-based pulldowns were enriched for RNA-related GO terms. This comprises the multiple molecular function (MF) GO terms connected to RNA binding or nucleic acid binding which are among the top five hits of significantly enriched GO terms in all three Csy4*-based pulldowns (Figure 2E, Appendix A). Strikingly, both interactors and candidates of all pulldowns have between 40 and 55% RNA-binders based on molecular function GO term annotations (Figure 2D). In addition, we have compared the list of all co-purified proteins with current in vivo mRNA interactome data and we can show that there is evidence for direct in vivo RNA binding for almost 50% of all identified proteins (Appendix A) [56,57,58,59]. Similarly, the biological process and cellular component GO terms are linked to a range of processes and components typically associated with RNA, such as RNA splicing, mRNA processing, the ribonucleoprotein complex or the spliceosomal complex (Appendix A). An exception to this are the co-purified proteins of the His_6_-MBP-Csy4*- and His_6_-TwinStrep-Csy4*-based pulldowns which are enriched for RNA-related molecular function GO terms, but show a tendency towards biological processes connected to photosynthesis (Appendix A). This is similar to a strong enrichment of metabolic and photosynthesis-related GO terms in our control samples (Appendix A). This is in line with a lower elution efficiency of the RNA bait in both pulldowns compared to the His_6_-Myc-Csy4*. Additionally, it is reasonable to find RNA-related proteins among our negative control samples, as we use Csy4*–RNA-loaded beads as negative controls.

Taken together, these results show that our modified Csy4*-based approaches for the identification of pri-miRNA stem-loop interactors successfully re-identified the known regulators of miRNA biogenesis and enrichment for proteins known to be involved in RNA binding. This pool of co-purified proteins represents a valuable source for the identification of new potential regulators in miRNA biogenesis.

### 2.5. Identification of New Potential Regulators of miRNA Biogenesis

To identify novel proteins with a potential regulatory function in miRNA biogenesis, we first selected all co-purified proteins containing a classical RNA binding domain (RBD) (Figure 3A, Appendix A). Strikingly, classical RBDs are among the most abundant protein domains identified for the co-purified proteins in each Csy4*-based pulldown (Appendix A). In total, we identified more than 100 different co-purified proteins containing at least one classical RBD. Of those which appear in at least one Csy4*-based pulldown, 33 proteins contained an RNA recognition domain (RRM), which consists of four antiparallel β-sheets and two α-helices and is one of the most abundant RBDs in Arabidopsis [60]. Thirteen proteins contained a WD40 domain, a conserved forty amino acid motifs with four to sixteen repeating units which often terminated in a tryptophan-aspartic (WD) dipeptide [61]. In addition to eleven DEAD-box domain-containing proteins, we also found six proteins with an S1 domain. DEAD-box helicases are a class of proteins which are able to unwind nucleic acids [62]. The S1 domain occurs in a wide range of RNA-associated proteins and its structure is similar to cold shock proteins, which bind nucleic acids [63].

Amongst the many RRM domain-containing proteins, the RRM and zinc-finger-containing protein *At*RZ-1a was identified in all three pulldowns and was, therefore, chosen for further functional analysis (Figure 1E–G and Figure 3B). As with the known pri-miR398b regulatory factor *At*GRP7, *At*RZ-1a is a family member of the Arabidopsis glycine-rich proteins (GRPs) and is located in both the nucleus and cytoplasm [64,65]. Moreover, both miR398 and *At*RZ-1a are involved in the response to abiotic stress, most notably oxidative stress. The level of miR398 is decreased upon induced oxidative stress, leading to the upregulation of the copper/zinc superoxide dismutase targets *CSD1* and *CSD2*, which are reactive oxygen species (ROS) scavengers [66]. In plants overexpressing *At*RZ-1a, H_2_O_2_ accumulated to higher levels compared to WT under stress conditions, but was lower in the mutants. Additionally, proteome analysis revealed that genes involved in ROS homeostasis were affected by *At*RZ-1a under stress [67].

To investigate this potential functional link and determine whether the interaction of *At*RZ-1a with the pri-miR398b stem-loop is reflected in altered miRNA processing, we analyzed miR398 steady-state levels in the *atrz-1a* mutant by small RNA blot analysis. We observed a decrease in the miR398 level in the *atrz-1a* mutant, in which miR398 was reduced to 60–70% of the wild-type levels (Figure 4A). Next, we determined whether *At*RZ-1a influences the pri-miR398a, b and c contents by analyzing the steady-state levels of all three precursors by quantitative PCR. Consistent with the observation that the expression of the mature miR398 is slightly reduced in the *atrz-1a* mutant and *At*RZ-1a interacts with the stem-loop structure of pri-miR398b, the steady-state level of pri-miR398b in the *atrz-1a* mutant is reduced to 60–70% of the wild-type (Figure 4B). In contrast, both pri-miR398a and c were unaffected in the *atrz-1a* mutant. This suggests that *At*RZ-1a might be a specific interactor of pri-miR398b and that *At*RZ-1a might specifically affect pri-miR398b stability. However, at this point we cannot exclude a transcriptional effect of *At*RZ-1a on pri-miRNA expression, as previously shown for SMA1, which plays a dual regulatory function in the transcriptional control and processing of pri-miRNAs [49]. Likewise, the possibility of indirect binding mediated by other protein interactors cannot be ruled out. Therefore, in vivo experiments such as iCLIP (individual nucleotide crosslinking and immunoprecipitation) combined with in vitro assays (e.g., EMSAs) will be important in the future to provide more mechanistic insights into the mode of interaction between pri-miR398b and *At*RZ-1a [29].

Another class of co-purified proteins are members of the RNA helicase superfamily (Figure 3A,B). In Arabidopsis, more than 120 genes were bioinformatically identified as potential members of this protein family within the Arabidopsis genome [62,68]. Among those are the RNA helicases of the DEAD-box family and related families. RNA helicases are enzymes that unwind double-stranded RNAs in an energy-dependent manner. As specific structural features are crucial for the proper processing of pri-miRNAs, RNA helicases may play a vital role in miRNA biogenesis [36,69,70]. This idea is further supported by the recent observation that the chromatin remodeling factor CHR2, the ATPase subunit of the large switch/sucrose non-fermentable (SWI/SNF) complex, directly binds to pri-miRNA stem-loops and remodels the secondary structure by its helicase activity [71]. Thereby, CHR2 inhibits the interaction of the processing machinery with the miRNA precursor and represses the accumulation of mature miRNAs. Among the co-purified proteins in our Csy4*-based pulldowns, we identified 11 RNA helicase superfamily proteins which contain at least one DEAD-box domain (Figure 3A,B). Of those, RNA HELICASE (RH) 11 and RH37 belong to the same clade of DEAD-box RNA helicases. Both proteins specifically bind to a specific set of small RNAs (sRNAs), including miRNAs, and function in the selective loading of sRNAs into extracellular vesicles for secretion [72]. Strikingly, miR398b was found to be localized in extracellular vesicles and delivered to the fungal pathogen *Botrytis cinerea* [73]. This suggests the possibility that the binding of RH11 and RH37 to miR398b is already initiated during the processing of pri-miR398b into the mature miR398b. With RH27, another RNA helicase required for zygote division and stem-cell homeostasis was already shown to bind to pri-miRNAs in vivo [74]. Furthermore, RH27 interacts with the processing components DCL1, DDL, HYL1 and SE in the nucleus. While the strong mutant allele *rh27-1* is zygote-lethal, the expression of several mature miRNAs, including miR398, is slightly reduced in the partial loss-of-function mutant *rh27-2*. This is accompanied by a reduced level of the corresponding pri-miRNAs. Importantly, the influence of RH27 on stem-cell-related genes is due to the reduced abundance of their regulatory miRNAs. In conclusion, it is likely that the identified plant RNA helicases regulate the processing of pri-miRNAs by remodeling their secondary structure.

Recent data already point to an intimate relation between the splicing machinery and pri-miRNA processing [75]. Several proteins we identified in our RNA pulldowns such as SE, *At*GRP7, SMA1 and members of the MAC have been shown to participate both in pre-mRNA splicing and miRNA biogenesis [15,40,46,49,76]. Beyond this, we have identified additional proteins as interactors of the pri-miR398b-SL for which a functional role in pre-mRNA splicing has been described. Among the co-purified proteins, we found the RRM-containing U1 SMALL NUCLEAR RIBONUCLEOPROTIN-70K and the SPLICEOSOMAL PROTEIN U1A which are both components of the U1 small ribonucleoprotein complex (U1 snRNP) that recognize the 5′ splice site during the splicing of mRNAs [77]. Note, it has been previously shown that pri-miRNA introns located downstream of the pri-miRNA stem-loop structure positively affect the biogenesis of miRNAs and that this effect depends on the 5′ splice site [78,79]. Interestingly, this regulatory mechanism is bidirectional meaning that miRNA processing and splicing machineries affect each other [79]. With the RNA HELICASE 2 (RH2) and the RRM protein Y14, we found two predicted members of the core exon junction complex which are colocalized in the nucleoplasm and are important factors for abiotic stress adaptation [80,81].

Additional splicing regulators, which we identified in our RNA pulldowns, are the LSM domain SM PROTEIN E1 (SME1) and the WD40 domain protein SUPPSSORS OF MEC-8 AND UNC-52 1 (SMU1). SME1 regulates spliceosome activity and, thus, the splicing of pre-mRNAs in a temperature-dependent manner [82,83]. Similar to SME1, SMU1 shows that the altered splicing of pre-mRNAs and absence of SMU1 causes developmental defects [84,85]. Although these splicing regulators have not been associated with miRNA biogenesis so far, the evident bidirectional regulation between both miRNA biogenesis and pre-mRNA splicing implies a putative regulatory function of these proteins in miRNA biogenesis.

In addition to co-purified proteins containing a classical RBD, several proteins with a role in central metabolism, which are not previously known to have a role in RNA-related processes, were identified in the three Csy4*-based pulldowns (Appendix A). These proteins may indicate a degree of unspecific contamination in the Csy4*-based RNA pulldowns, as many of them are highly expressed in plants. However, metabolic proteins which are significantly enriched in at least one Csy4*-based pulldown could also have functional significance. Fructose 1,6-bisphosphate aldolase 6 (FBA6) which is involved in glycolysis and the Calvin cycle is among the significantly enriched interactors in His_6_-MBP-Csy4*- and His_6_-TwinStrep-Csy4*-based pulldowns and among the candidates in the His_6_-Myc-Csy4*-based pulldown, for example [86]. Strikingly, two independent mRNA interactome studies provide evidence for the direct binding of FBA6 to mRNAs in vivo [56,57]. A similar observation concerning the appearance of metabolic enzymes were made in a high-throughput screen for pre-miRNA binding proteins in human cell lines [87]. Independent confirmation of the pri-miRNA binding activity would then point to a double-life of metabolic enzymes in miRNA biogenesis.

## 3. Materials and Methods

### 3.1. Cloning and In Vitro Transcription

DNA fragments corresponding to the pri-miR398b stem-loop structure including the 20-nt Csy4-hairpin, T7 promoter and spacer sequences were synthesized as a gBlock^TM^ (IDT, Coralville, IA, USA) and subcloned into the pUC19 vector via *Eco*RI and *Bam*HI restriction sites (Appendix A). The vector was linearized with *Hind*III and used as template for in vitro transcription. Csy4-hairpin-tagged pri-miR398b stem-loops were in vitro transcribed using the MEGAshortscript^TM^ T7 transcription kit (Invitrogen, Carlsbad, CA, USA) according to the manufacturer’s instructions. The in vitro transcript was purified using the RNA Clean & Concentrator-25 Kit (Zymo Research, Freiburg, Germany).

### 3.2. Cloning and Recombinant Protein Expression of Tagged Csy4* Protein Variants

For recombinant expression of His_6_-MBP-Csy4* in *Escherichia coli* BL21 DE3, we used the original Gateway-based pHM-GWA_Csy4H29AS50C (Addgene plasmid #45029) expression vector [21]. To produce the His_6_-TwinStrep- and His_6_-Myc-tagged Csy4*, we replaced the His_6_-MBP coding sequence in pHM-GWA_Csy4H29AS50C by synthetic gBlock^TM^ fragments (IDT) containing sequences for His_6_-TwinStrep and His_6_-Myc via *Xba*I and *Not*I restriction sites and verified the constructs through sequencing. gBlock^TM^ sequences are listed in Appendix A. For recombinant expression of His_6_-TwinStrep-Csy4* and His_6_-Myc-Csy4*, the modified pHM-GWA_Csy4H29AS50C vectors were transformed into *E. coli* BL21 DE3 or into Tuner^TM^ DE3 cells (Novagen, Darmstadt, Germany), respectively. Protein expression was induced with 0.5 mM isopropyl β-D-1 thiogalactopyranoside (IPTG) at an optical density (OD_600_) of 0.5, followed by shaking at 18 °C for ~16 h. Cells were resuspended in low imidazole buffer (15.5 mM Na_2_HPO_4_, 4.5 mM NaH_2_PO_4_, pH 7.5, 150 mM NaCl, 10 mM imidazole, 0.01% Triton X-100, 5% glycerol, 1 mM ß-mercaptoethanol) and disrupted by passing three times through a French pressure cell at 1000 psig. The clarified lysates were incubated with PureCube Indigo Ni-NTA affinity resin (Cube Biotech, Monheim, Germany) for 1.5 h at 4 °C under moderate shaking. Subsequently, the resins were washed several times with low imidazole buffer (15.5 mM Na_2_HPO_4_, 4.5 mM NaH_2_PO_4_, pH 7.5, 150–500 mM NaCl, 10–100 mM imidazole, 0.01% Triton X-100, 5% glycerol, 1 mM ß-mercaptoethanol) containing variable amounts of salt and imidazole, as indicated in Appendix A. After washing, the bound proteins were eluted in 3–5 fractions with high imidazole buffer (15.5 mM Na_2_HPO_4_, 4.5 mM NaH_2_PO_4_, pH 7.5, 300–500 mM NaCl, 150–300 mM imidazole, 0.01% Triton X-100, 5% glycerol, 1 mM ß-mercaptoethanol) for 1 h at 4 °C under moderate shaking, as indicated in Appendix A For further use of the protein in Csy4*-based RNA pulldowns, high imidazole buffer was exchanged with low imidazole buffer by centrifugation through Amicon Ultracel-30 Unit (30 kDa MWCO) or Ultracel-10 Unit (10 kDa MWCO) Centrifugal Filter.

To check for protein stability, differential scanning fluorimetry (DSF) was carried out essentially, as described previously [88]. Briefly, proteins were measured at 0.1 mg/mL with 1x SYPRO Orange from 26 °C to 95 °C. Buffers were based on potassium phosphate, 2-Hydroxyethylpiperazine-N’-2-ethanesulfonic acid (Hepes), ammonium acetate, sodium phosphate, Tris-(hydroxymethyl)-aminomethan (Tris), imidazole and N,N-Bis(2-hydroxyethyl)glycine (Bicine). The pH of the tested buffers ranged from 7.0 to 9.0 while NaCl concentration increased from 20 mM to 1 M. Data were analyzed as previously described [34]. Protein concentrations were determined using the Qubit^TM^ Protein Assay Kit (Thermo Fisher Scientific, Waltham, MA, USA) and adjusted to 10 mg/mL.

### 3.3. Nucleoplasmic Extract Preparation

Five grams of Arabidopsis Col-0 wild-type plant material was ground in liquid nitrogen and resuspended in 15 mL nuclear lysis buffer (20 mM Tris-HCl, pH 7.4, 25% glycerol, 20 mM KCl, 2 mM Na_2_-EDTA, pH 8.0, 2.5 mM MgCl_2_, 250 mM sucrose, 5 mM dithiothreitol (DTT)). All following steps were performed on ice or at 4 °C. The samples plus 5 mL nuclear lysis buffer used for rinsing was successively filtered through a 100 µm nylon filter and two layers of Miracloth (Calbiochem; 22–25 µm), resulting in the total lysate. The sample was centrifuged at 1500× *g* for 30 min to separate the nuclei and the cytosolic fraction. The supernatant was transferred into a fresh tube and corresponds to the cytosolic fraction. The pellet from the 1500× *g* centrifugation step was resuspended in 12 mL nuclei resuspension buffer (20 mM Tris-HCl, pH 7.4, 25% glycerol, 2.5 mM MgCl_2_, 0.2% Triton X-100) and centrifuged again at 1500× *g* for 15 min. This washing step was repeated four-to-five times until the supernatant became clear. Finally, the pellet was suspended in 3 volumes (~600 µL) of protein extraction buffer (20 mM Hepes-KOH, pH 7.5, 5% glycerol, 1.5 mM MgCl_2_, 0.2 mM Na_2_-EDTA, pH 8.0, 420 mM NaCl, 5 mM DTT, 1x Complete Protease inhibitor (Roche, San Francisco, CA, USA), as previously described [89]. For Csy4*-based RNA pulldowns, the final salt concentration was adjusted to 150 mM NaCl. Protein concentrations were determined using the Qubit^TM^ Protein Assay Kit (Thermo Fisher Scientific).

### 3.4. Immunoblot Analysis

Immunoblot analysis of total lysate, cytoplasmic and nucleoplasmic fractions was essentially performed as described [29]. Briefly, primary antibodies were polyclonal antibodies against HISTONE 3 (Agrisera, Vännäs, Sweden, AS10710; rabbit; dilution 1:5000), PHOSPHOENOLPYRUVATE CARBOXYLASE (Agrisera AS09458; rabbit; dilution 1:20,000) and SERRATE (Agrisera AS09532A; rabbit; dilution 1:1000). Secondary antibody was HRP-coupled anti-rabbit IgG (Sigma-Aldrich, Burlington, MA, USA, A0545; dilution 1:2500). Chemiluminescence detection of the immunoblots was done with Fusion-FX6 (Vilber) system.

### 3.5. Csy4*-Based RNA Pulldowns

A total of 20 µL MBP-Trap^®^ agarose beads, 20 µL Myc-Trap^®^ agarose beads (Chromotek, Martinsried, Germany) and 10 µL StrepTactin^®^ XT agarose beads (IBA life sciences, Göttingen, Germany) were washed three times in 150 µL 1x RNA binding buffer (20 mM Tris-HCl, pH 7.5, 100 mM NaCl, 5% glycerol, 1 mM Tris(2-carboxyethyl)phosphine hydrochloride (TCEP), 0.1% Triton X-100), respectively. To allow protein coupling to the beads, 40 µg of purified His_6_-MBP-Csy4*, His_6_-TwinStrep-Csy4* or His_6_-Myc-Csy4* protein was added to either washed MBP-Trap^®^, Myc-Trap^®^ or StrepTactin^®^ XT agarose beads and incubated for 1 h at 4 °C with constant shaking in 250 µL RNA binding buffer. After coupling, the beads were washed 3×with 300 µL RNA binding buffer. To allow correct folding of the RNA, 2.75 µg of in vitro-transcribed Csy4-hairpin-tagged pri-miR398b stem-loops were diluted in 250 µL RNA folding buffer (20 mM Tris-HCl, pH 7.5, 100 mM, 3 mM MgCl_2_), heated to 70 °C for 10 min, placed on ice for 2 min and incubated at 20 °C for 10 min. The RNA was then added to the respective Csy4*-coupled beads and incubated for 90 min at 4 °C with constant shaking. The beads were again washed 3× with 150 µL RNA binding buffer. The nucleoplasmic extract was added to the beads and incubated for 2 h at 4 °C under constant rotation. The beads were washed 3× 15 min with 500 µL RNA binding buffer. Cleavage and elution of the RNA was induced by addition of 50 µL elution buffer (20 mM Tris-HCl, pH 7.5, 100 mM NaCl, 5% glycerol, 1 mM TCEP, 500 mM imidazole) and incubation at 4 °C for 1 h. Control samples were treated with elution buffer omitting the imidazole. The supernatant was collected and NuPAGE^TM^ LDS sample buffer (4×) (Thermo) was added to prepare protein samples for denaturing gel electrophoresis, according to the manufacturer’s instructions. In vitro cleavage assays were essentially performed as described above, but without the addition of nucleoplasmic extract. Next, 50 µL of 2× loading buffer (90% (*w*/*v*) formamide, 9.8% (*w*/*v*) glycerol, 0.1% (*w*/*v*) bromphenol blue, 0.1% (*w*/*v*) xylene cyanol) were added to 50 µL of each fraction and boiled at 90 °C for 3 min. Then, 20% of each fraction was loaded on 10% polyacrylamide-8M urea gel in Tris-borate-EDTA buffer and visualized by SYBR^TM^ Gold (Thermo) staining.

### 3.6. Mass Spectrometry and Data Analysis

Protein samples were separated on a 4–12% NuPAGE^TM^ Novex Bis-Tris precast gel (Thermo) for 8 min at 180 V in 1× MES buffer (Thermo). After protein fixation and Coomassie blue staining (Carl Roth, Karlsruhe, Germany), gels were minced and destained in 50% EtOH/25 mM ammonium bicarbonate (ABC) followed by dehydration in 100% acetonitrile (ACN). Subsequently, samples were reduced for 1 h in 10 mM DTT/50 mM ABC (Sigma-Aldrich) at 56 °C followed by alkylation with 50 mM iodoacetamide/50 mM ABC (Sigma-Aldrich) for 45 min at room temperature in the dark. Gel pieces were dehydrated with pure acetonitrile (VWR) and afterwards incubated with 1 μg mass-spec grade trypsin (Serva) at 37 °C overnight. Tryptic peptides were extracted twice with 30% acetonitrile and twice with pure acetonitrile.

The mixture with the extracted peptides was concentrated in a concentrator (Eppendorf) to a final volume of ca. 100 µL. Peptides were desalted on StageTips [90] and separated on a nanocapillary (New Objective) packed in-house with Reprosil C18 (Dr. Maisch GmbH, Ammerbuch, Germany). The column was attached to an Easy nLC 1000 system (Thermo) operated with a 90 min gradient from 5% to 60% acetonitrile in 0.1% formic acid at a flow of 225 nL/min. The spray capillary was mounted on the nanospray ion source of a Q Exactive Plus mass spectrometer (Thermo). Measurements were using HCD fragmentation with a data-dependent Top10 MS/MS spectra acquisition scheme per MS full scan in the orbitrap analyzer.

The raw files were processed with MaxQuant (version 1.5.2.8, [91]) and searched against the TAIR10 protein database (35,386 entries). MaxQuant standard settings were used, except label-free quantitation [92], without FastLFQ and match between runs option was activated. Prior to statistical analysis, entries from contaminants, reverse hits and only identified by site were removed. Missing values were imputed by a downshifted compressed beta distribution at the limit of quantitation. Enrichment was calculated as the mean of the 4 replicates and the variation was statistically assessed by Welch *t*-test using R. Logarithmic transformations of the enrichment were plotted against the determined *p*-value. Proteins with a *p*-value ≤ 0.05 and a log_2_ fold change ≥ 1.0 were considered as significantly enriched.

### 3.7. Analysis of GO Terms and Protein Domains

Genes enriched in gene ontology (GO) terms were analyzed in the online Thalemine database using the default background population and Holm–Bonferroni test correction [93]. Protein domains, including classical and non-classical RNA binding domains, were selected from all Pfam domains by manual annotation, as previously described [58].

### 3.8. Plant Material and Growth Conditions

Arabidopsis seeds were surface-sterilized and sown on half-strength MS (Murahige-Skoog; Duchefa, Haarlem, Netherlands) plates [94]. Plants were grown in 12 h light–12 h dark cycles at 20 °C in Percival incubators (CLF, Wertingen, Germany). For preparation of nucleoplasmic extracts and RNA analysis, 14-day-old aerial tissue from Col-0 wild-type was harvested at dusk. The *atrz-1a* T-DNA mutant (SALK_036865C) has been described previously [65]. Col-0 served as a wild-type control in all experiments.

### 3.9. Detection of Small RNAs by Northern Blotting

For RNA analysis, at least ten plants were bulked for each sample per replicate and RNA was isolated using TRIzol^®^ reagent (Invitrogen). For small RNA gel blots, 5 µg of total RNA was fractionated on a 17% polyacrylamide-8M urea gel in Tris-borate-EDTA buffer, transferred onto GeneScreen membrane by capillary blotting and crosslinked with UV-light (254 nm) at 120 mJ/cm^2^. Hybridization was performed with 5′-biotinylated antisense DNA-oligonucleotides (Metabion, Planegg, Germany) in hybridisation buffer (7% SDS, 200 mM Na_2_HPO_4_, pH 7, 0, 5 µg/mL salmon sperm DNA). Hybridization was performed for 12–15 h at 38 °C with end-over-end rotation and subsequently rinsed with washing buffer (1× SSC, 0.1% SDS) twice for 15 min at 38 °C. The biotin-labeled probes were detected using the Nucleic Acid Detection Module Kit (Thermo Fisher Scientific), according to the manufacturer’s instructions. Chemiluminescence detection was done with Stella (Raytest) and signals were quantified using ImageJ (v. 1.53, Rasband, W.S., NIH, Bethesda, MD, USA) [95]. A Student’s *t*-test was performed to determine statistical significance (** *p* ≤ 0.05, * *p* ≤ 0.1).

### 3.10. RT-qPCR

For RT-qPCR, RNA was reverse transcribed using AMV Reverse transcriptase (EUR_X_ Roboklon, Berlin, Germany). qPCR was performed in a volume of 10 µL with the iTaq SYBR GREEN supermix (Biorad, Hercules, CA, USA) using 45 cycles of 15s at 95 °C and 30 s at 60 °C in a CFX96 cycler (Biorad). C_q_ values were determined and relative expression levels were calculated based on non-equal efficiencies for each primer pair [96]. Data were normalized to *PP2A* (At1g13320) and expressed as mean xpression levels of the independent biological replicates each ± standard deviation or as indicated in the figure legend. A Student’s *t*-test was performed to determine statistical significance (** *p* ≤ 0.05, * *p* ≤ 0.1). Primers are listed in Appendix A.

## 4. Conclusions

Accumulating evidence indicates that transcription is not an all-determining force that controls the expression of plant miRNAs. The complex and highly variable secondary structure of plant pri-miRNA stem-loops constitutes the basis for extensive regulation by RBPs on the RNA level. However, it is still unclear as to what extent particular pri-miRNA stem-loops are recognized by specific regulatory RBPs.

Here, we successfully developed three simplified versions of the original Csy4*-based RNA-pulldown for the identification of specific pri-miRNA stem-loop interacting proteins, which are easily applicable for routine use in molecular biology laboratories. Comparing the three approaches, the Csy4* protein with the Myc tag showed the best overall performance, as it both efficient coupling to beads and the specific cleavage and elution of the pri-miR398b stem-loop was achieved, which is key for identifying high-confidence protein interactors. The Myc-tag also has the advantage of being small in size and, therefore, having minimal effects on the structure and activity of Csy4*.

Using these modified protocols, we identified known processing factors and dozens of potentially new regulators in miR398 biogenesis. Among those, we succeeded in characterizing the glycine-rich RBP *At*RZ-1a as a novel regulator of pri-miR398b processing. The absence of *At*RZ-1a caused a reduction in the steady-state abundance of the mature miR398 and its pri-miR398b precursor suggesting that *At*RZ-1a positively affects the biogenesis of the mature miR398. Although the effect observed under standard growth conditions is rather mild and the exact role of *At*RZ-1a in regulating pri-miR398 is yet to be determined, it is possible that the extent of this regulation becomes more pronounced under stress conditions, as both *At*RZ-1a and miR398 are involved in the response to abiotic stress [65,66,67]. In the future, it will be interesting to investigate whether *At*RZ-1a specifically acts on pri-miR398b or whether it has a global impact on miRNA processing, in particular, in the context of the stress response.

In conclusion, our Csy4*-based in vitro approaches allow for the simple and rapid identification of pri-miRNA stem-loop interactors with a functional role in plant miRNA biogenesis. Nevertheless, further optimizations of the method, such as improving the elution of the RNA bait together with bound interactors and extended washing steps to reduce the background, show great potential to improve the stringency and identification of high-confidence interactors.

For a more realistic scenario under physiological conditions, a future step would be the identification of specific RNA interactors in vivo. Techniques have already been established for mammalian cell cultures and are based on the specific purification of UV-crosslinked RNA–protein complexes using immobilized antisense oligonucleotides that are complementary to the RNA of interest [97,98,99]. However, these methods are technically challenging and were mainly used to identify interactors of highly expressed RNAs. Due to the low abundance of the short-lived pri-miRNAs in plants and the complex composition of plant cell extracts containing UV-absorbing pigments and large amounts of endogenous RNases, these elaborate techniques cannot simply be transferred to plants. For this reason, our modified Csy4*-based strategy provides a valuable state-of-the-art technique to identify specific *trans*-acting regulators of particular pri-miRNA stem-loops and may be extended to other classes of RNAs.

## Figures and Tables

**Figure 1 ijms-23-08961-f001:**
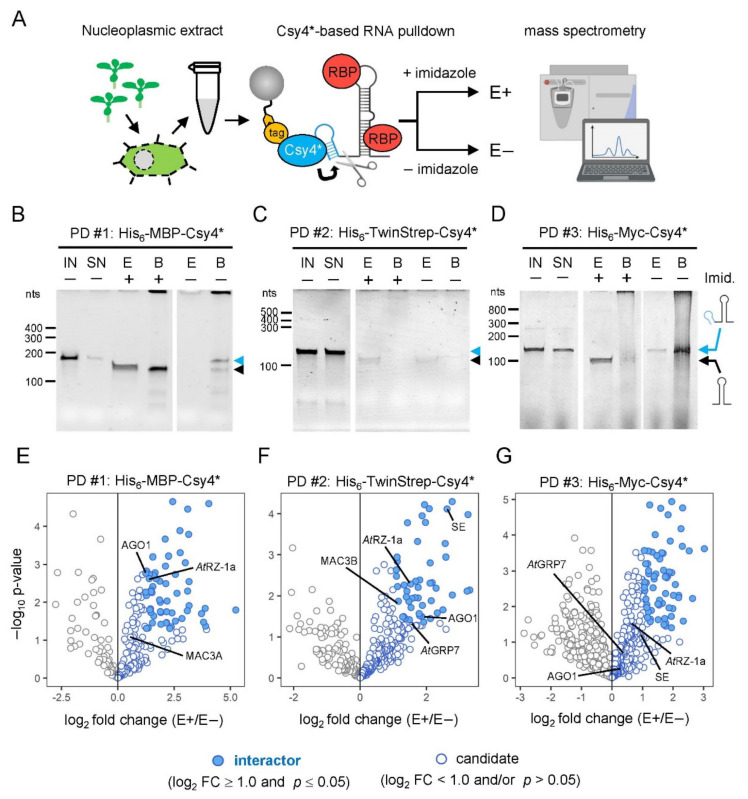
**Identification of specific pri-miR398b stem-loop interacting proteins.** (**A**) Schematic representation of the Csy4*-based RNA pulldown strategy to identify pri-miRNA stem-loop interacting proteins. The scissors indicate cleavage activity upon imidazole supply. RBP, RNA-binding protein. (**B**–**D**) In vitro cleavage activity of His_6_-MBP-Csy4* (**B**), His_6_-TwinStrep-Csy4* (**C**) and His_6_-Myc-Csy4* (**D**). Cleavage products were separated on denaturing urea-PAGE gels and visualized with SYBR^TM^ Gold staining. IN, input RNA. SN, supernatant after RNA coupling. E+, eluted fraction with imidazole. E−, control fraction without imidazole. B+, remaining beads after elution with imidazole. B−, remaining beads without imidazole elution. Imid., imidazole. (**E**–**G**) Volcano plots showing the proteins co-purified with the pri-miR398b stem-loop using either His_6_-MBP-Csy4* (**E**), His_6_-TwinStrep-Csy4* (**F**) or His_6_-Myc-Csy4* (**G**). Proteins with a *p*-value ≤ 0.05 and a log_2_ fold change (FC) ≥ 1.0 are defined as interactors. Proteins with a *p*-value ˃ 0.05 and/or a log_2_ FC ˂ 1.0 are defined as candidates. AGO1: ARGONAUTE 1. MAC3A and 3B: MOS4-ASSOCIATED COMPLEX 3A and 3B. *At*GRP7: *Arabidopsis thaliana* GLYCINE-RICH PROTEIN 7. SE: SERRATE.

**Figure 2 ijms-23-08961-f002:**
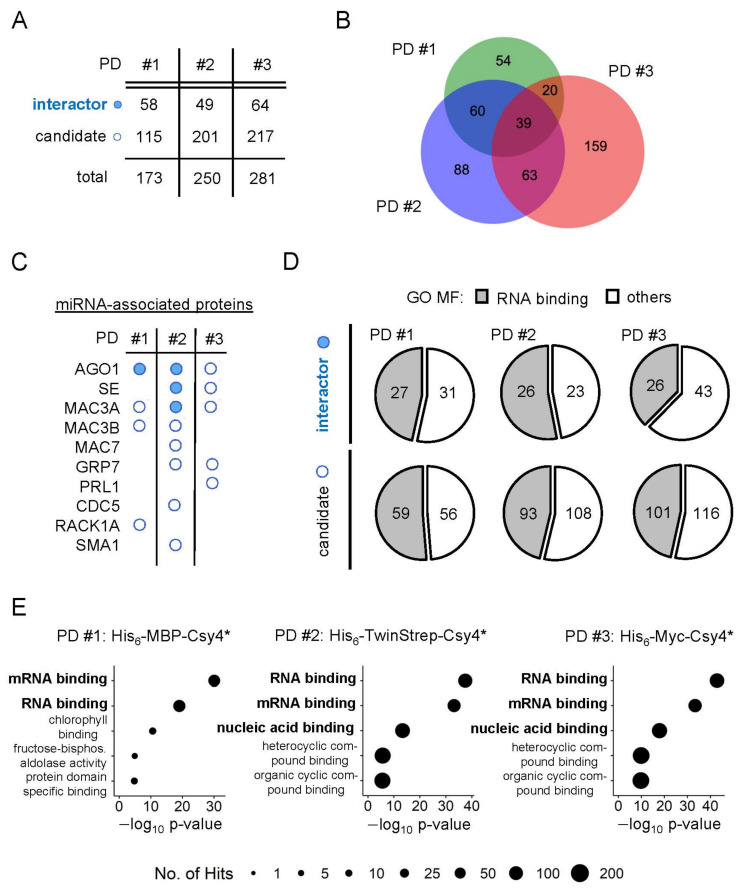
**In silico analysis of co-purified proteins.** (**A**) Number of co-purifying proteins which were classified as interactors or candidates. (**B**) Overlap of co-purified proteins (combined interactors and candidates) between the three Csy4*-based RNA pulldowns. (**C**) Co-purified proteins already known to be associated with miRNA biogenesis. (**D**) Proportion of interactors and candidates of each pulldown that are associated with RNA-binding related molecular function (MF) GO terms. (**E**) Top 5 enriched GO MF terms of all interactors and candidates. PD #1: His_6_-MBP-Csy4*, PD #2: His_6_-TwinStrep-Csy4*, PD #3: His_6_-Myc-Csy4*.

**Figure 3 ijms-23-08961-f003:**
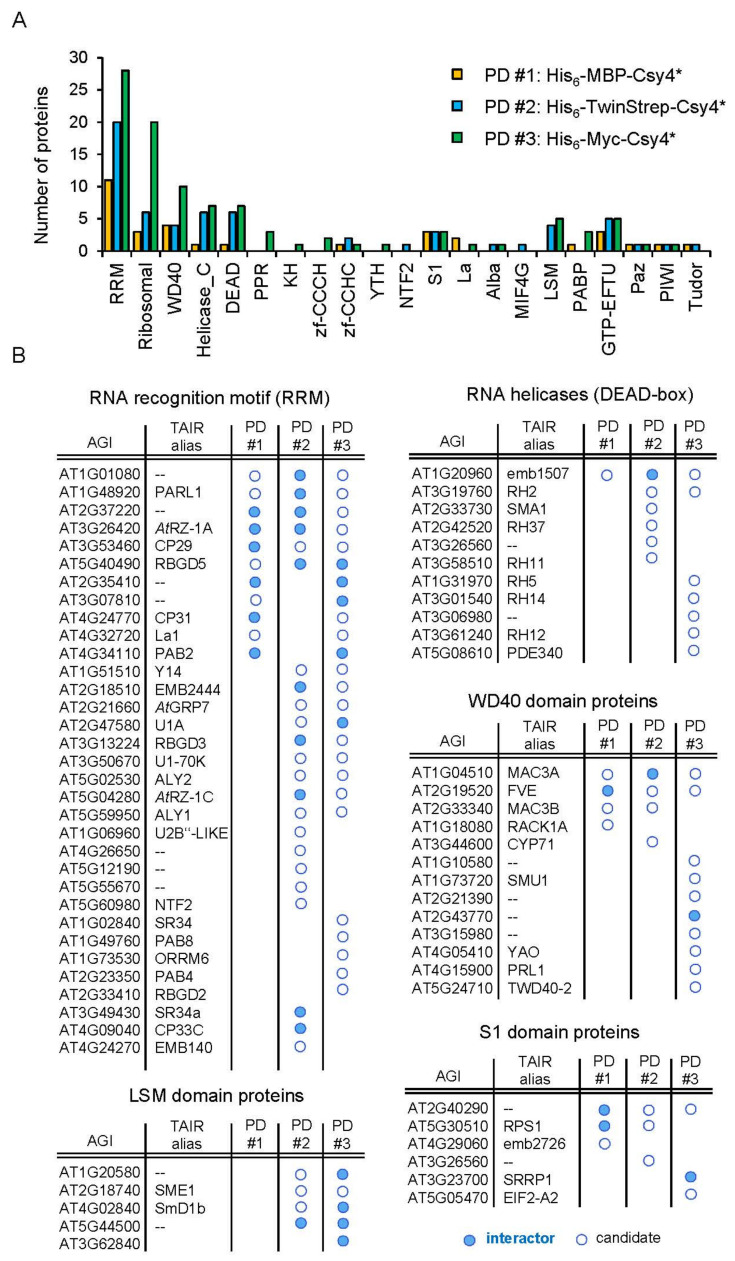
**Systematic categorization of co-purified proteins.** (**A**) Number of proteins harboring domains associated with RNA-binding in RNA pulldowns using His_6_-MBP-Csy4* (yellow), His_6_-TwinStrep-Csy4* (blue) and His_6_-Myc-Csy4* (green). (**B**) Overview of co-purified proteins containing a recognized RNA-binding domain. Shown are interactors and candidates containing an RNA recognition motif (RRM), a LSM domain, a DEAD-box helicase domain, a WD40 domain and/or S1 domain.

**Figure 4 ijms-23-08961-f004:**
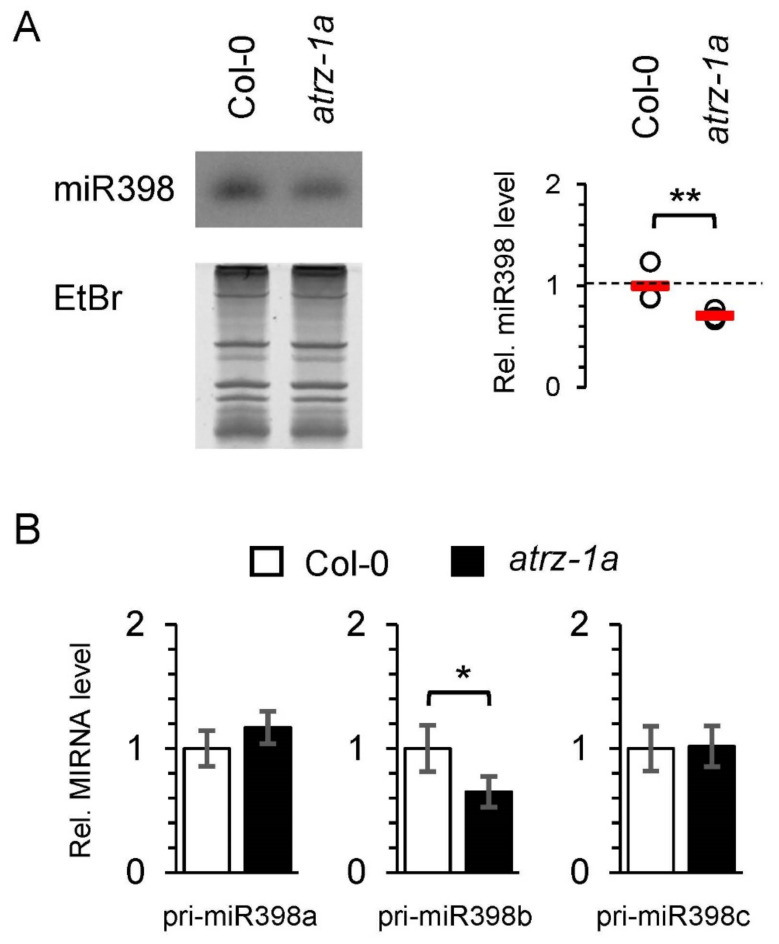
***At*RZ-1a affects the biogenesis of miR398.** (**A**) MiR398 levels are reduced in *atrz-1a* mutants. An RNA gel blot of the *atrz-1a* mutant and the corresponding wild-type plants was hybridized with an anti-miR398 probe. Ethidium bromide (EtBr) staining serves as a loading control. MiR398 signals were quantified using ImageJ and fold changes of miR398 are expressed relative to wild-type. Shown are the mean ± SD of three biological replicates. (**B**) Pri-miR398b level is decreased in *atrz-1a* mutants. The levels of the pri-miR398a, b and c were analyzed in the *atrz-1a* mutant and wild-type by quantitative real-time PCR. Data are based on three biological replicates and shown as mean ± SD. A Student’s *t*-test was performed to determine statistical significance (** *p* ≤ 0.05, * *p* ≤ 0.1).

## Data Availability

The data presented in this study are available in Appendix A.

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
