# Peer review of "Identification of Pri-miRNA Stem-Loop Interacting Proteins in Plants Using a Modified Version of the Csy4 CRISPR Endonuclease"

_ijms, 2022, doi:10.3390/ijms23168961_

Round 1

Reviewer 1 Report

The experimental methods and the information they obtained seem very powerful and useful, and every steps of the experiments were carefully pre-checked and performed. Thus, I believe that this manuscript deserves to be published in the IJMS journal. Though, some typos and simple errors were found in several places, so it is recommended that the manuscript be carefully re-examined.

Greek symbols ‘alpha’ and ‘beta’ appear to be missing in line #579. In lines #622 and #626, a numerical symbol (±) seems missing in ‘mean _ SD’. Hyphens appear to be needed in several places, for instance line #351 (‘tag containing’ to ‘tag-containing’), line #560 (‘photosynthesis related’ to ‘photosynthesis-related’), line #582 (‘domain containing’ to ‘domain-containing’), and line #664 (‘RRM containing’ to ‘RRM-containing’).

Reviewer 2 Report

RNA-binding proteins and their role in regulating the expression of microRNAs in plants is an emerging field. Inspired by a 2013 study in animal cells using CRISPR nuclease, Lüders and co-workers designed a simple way of RNA pulldown strategy, which has great potential in identifying novel proteins that bind to the pri-miRNA stem-loop and may provide novel insights into the post-transcriptional regulation of plant miRNAs. The experiments were carefully performed and the manuscript is well-written. I only have the following minor comments that may improve the manuscript:

1.    In Figure 4, the authors show that the atrz-1a mutant has about 70% of miR398 as that of Col-0 accumulated. In addition, relative MIRNA levels of pri-miR398b are only slightly reduced in atrz-1a mutant. I am wondering if this is sufficient to hypothesize the role of AtRZ-1a in controlling miR398. Even though the arguments in lines 612-613 per se are valid, it would be more appropriate to assume that AtRZ-1a has only a minor role in regulating miR398. I think it is worth mentioning this point somewhere in the text. In addition, to unequivocally prove the physical interaction between AtRZ-1a and pri-miR398, an in vitro EMSA assay may be performed.

2.    Also, it is possible that AtRZ-1a may affect the stability of miRNAs other than 398. One way to check this would be to perform expression analysis of many known miRNAs via stem-loop qPCR or small RNA blots using atrz-1a mutant and Col-0 plants. However, this may be out of the purview of this manuscript. At least these possibilities should be discussed.

3.    The three different pull-down experiments using various tags are well thought-out. The authors also provide a comparison among these three methods in terms of overall protein produced, binding affinity as well as differences in coupling and elution. However, this information is scattered throughout the results section. It would be helpful if the authors could consolidate the comparisons and write a sentence or two in the discussion section. This would greatly facilitate the proper choice of pull-down method for future research

4.    Even though a number of proteins that passed the strict cut-off criteria used by the researchers, were present in all three pull-down experiments, the reason for choosing AtRZ-1a for functional analysis is not very clear (Lines 598-599). Would it be possible to highlight the known structural or functional features of AtTZ-1a to substantiate this further?

5.    In Figure legends and in the text, please make atrz-1a (that points to the mutant line) italicized

6.    In the legend of Figure 1, the addition of abbreviations for protein names (such as SE, AGO1 etc. in Figures 1E to G) would greatly help the readers

Reviewer 3 Report

Manuscript "Identification of pri-miRNA stem-loop interacting proteins in plants using a modified version of the Csy4 CRISPR endonuclease" is very interesting

General comments:

Authors developed a simplified version of the original Csy4* protocol which is easily applicable for routine use in molecular biology laboratories. Authors selected and engineered three different epitope-tagged versions of Csy4* and compared their performance in RNA pulldown experiments.

M&M section is described good but unfortunately lack is statistical analysis section.

Detailed comments:

Figure 1: Lack is statistical comparison between interactor and candidate.
Figure 2: Lack is statistical analysis of results presented in this figure.
Figure 4 needs LSD or HSD values.

Paper needs major revision.

Round 2

Reviewer 3 Report

Now, all is ok.